# Exploring the Ability of Solar-Induced Chlorophyll Fluorescence for Drought Monitoring Based on an Intelligent Irrigation Control System

**Wenhui Zhao** [1,2,3,4]**, Jianjun Wu** [1,2,3,4,*]**, Qiu Shen** [1,2,3,4]**, Jianhua Yang** [1,2,3,4] **and Xinyi Han** [1,2,3,4]

1   State Key Laboratory of Remote Sensing Science, Beijing Normal University, Beijing 100875, China
2   Beijing Key Laboratory for Remote Sensing of Environment and Digital Cities, Beijing 100875, China
3   Faculty of Geographical Science, Beijing Normal University, Beijing 100875, China
4   Center for Drought and Risk Research, Beijing Normal University, Beijing 100875, China
*   Correspondence: jjwu@bnu.edu.cn

**Abstract:** Drought is one of the most devastating disasters and a serious constraint on agricultural development. The reflectance-based vegetation indices (VIs), such as Normalized Difference Vegetation Index (NDVI), have been widely used for drought monitoring, but there is a lag in the response of VIs to the changes of photosynthesis induced by drought. Solar-induced chlorophyll fluorescence (SIF) is closely related to photosynthesis of vegetation and can capture changes induced by drought timely. This study investigated the capability of SIF for drought monitoring. An intelligent irrigation control system (IICS) utilizing the Internet of Things was designed and constructed. The soil moisture of the experiment plots was controlled at 60–80% (well-watered, T1), 50–60% (mild water stress, T2), 40–50% (moderate water stress, T3) and 30–40% (severe water stress, T4) of the field water capacity using the IICS based on data collected by soil moisture sensors. Meanwhile, SIF, NDVI, Normalized Difference Red Edge (NDRE) and Optimized Soil Adjusted Vegetation Index (OSAVI) were collected for a long time series using an automated spectral monitoring system. The differences in the responses of SIF, NDVI, NDRE and OSAVI to different drought intensities were fully analyzed. This study illustrates that the IICS can realize precise irrigation management strategies and the construction of regulated deficit irrigation treatments. SIF significantly decreased under mild stress, while NDVI, NDRE and OSAVI only significantly decreased under moderate and severe stress, indicating that SIF is more sensitive to drought. This study demonstrates the excellent ability of SIF for drought monitoring and lays the foundation for the future application of SIF in agricultural drought monitoring.

**Keywords:** drought; SIF; vegetation index; winter wheat

## 1. Introduction

In arid and semiarid areas such as the North China Plain, water is a scarce and valuable resource [1]. Agricultural irrigation accounts for 70% of irrigation-related withdrawals and is the largest freshwater consumption source in the world from a statistical analysis reported by the Food and Agriculture Organization of the United Nations [2,3]. Population growth and industrial development have led to intensified competition for water resources, resulting in fewer water resources for agricultural irrigation [4]. Therefore, rational planning of irrigated amounts and durations to improve the crop water use efficiency (WUE) has become increasingly important. One way to improve the WUE is to enhance the yield per unit of water used while reducing the waste of ineffective water resources [5,6]. The purpose of irrigation is to minimize water use and maximize crop yields [7,8]. The irrigation strategy determines the irrigation time and amount on the basis of crop types, growing periods and environmental conditions [9].

To implement irrigation strategies accurately, the levels of water stress must be clearly understood. According to the type of water stress, irrigation strategies can be implemented based on the soil water balance (WB), soil moisture monitoring methods, and plant water stress degrees [10]. The WB approach has been widely employed to determine irrigation requirements for various crops. In this method, the water input into the plant-soil system needs to be counterpoised with the output [11]. Crop evapotranspiration (ETc), which considers both soil evaporation and plant transpiration, is the most important component of the WB [12,13]. ETc is estimated by the product of the crop coefficient Kc and the reference crop evapotranspiration (ETo). The formula is expressed as ETc = Kc × ETo, where Kc estimates the amount of evapotranspiration by integrating the biophysical and physical differences between a reference plant and the target plant, while ETo expresses the requirements for crops under certain environmental conditions. The advantage of the WB-based irrigation method is that it can predict the water demand of crops at a specific growth stage, and the irrigation strategy can be reasonably adjusted according to this feature [14]. However, the accuracy of this method in terms of its practical applications is questioned, because even for specific crops, the value of Kc may vary due to planting density and direction [15], plant variety [16] and canopy structure [17]. In addition, it is difficult to accurately obtain the parameters of the WB method. Therefore, the simple irrigation strategy is based on soil moisture measurements. By setting upper and lower thresholds, irrigation begins when the soil moisture decreases to the lower threshold, and irrigation stops when the soil moisture rises to the upper threshold [18,19].

To address these concerns, a decision support system has been developed [20]. It is employed to help decision makers gather useful information to identify problems and formulate optimized strategies [21]. The simplest decision system implements automatic irrigation based on the soil volumetric water content data obtained from a real-time monitoring capacitive soil water sensor [22,23]. Irrigation is started or stopped when the sensor measurement value is below or above the predetermined threshold value [24,25]. Some researchers have used the decision support system to implement irrigation strategy which demonstrated regulated deficit irrigation could be achieved without human intervention [26]. However, one drawback of this approach is the lag in the soil moisture response. Since it takes time for the irrigated water to seep through the soil profile, the amount of irrigation may exceed the expected value by the time the soil moisture sensor reaches the upper limit set to terminate irrigation. Thus, the solution of this study is to improve the current decision support system by establishing upper and lower irrigation thresholds, calculating the amount of irrigation needed for soil moisture to reach the upper limit from the set lower limit, and automatically terminate the system processing when the irrigation reaches the set amount after initiation.

With the development of remote sensing technology, solar-induced fluorescence (SIF) offers unique opportunities for environmental stress (drought and high temperature) monitoring and estimation of gross primary productivity (GPP). Several studies have explored the application of SIF. For example, Lee et al. [27] used SIF and Enhanced Vegetation Index (EVI) to detect the effects of drought in Amazon forest, and they showed that SIF showed a significant decrease in the central Amazon forest during the dry season, while the change in EVI was not significant. Liu et al. [28] explored the response of SIF and the NDVI of winter wheat to different drought intensities. They found that SIF was significantly reduced under severe and extreme drought conditions, while the NDVI was significantly reduced only under extreme drought. Moreover, the results showed that SIF was significantly and positively correlated with soil moisture, indicating that SIF can capture agricultural drought information based on the value of soil moisture. The above results suggest that SIF is more suitable for agricultural drought monitoring. Song et al. [29] used SIF, the NDVI and EVI to monitor the effects of high temperature stress on wheat in Northwestern India. The results showed that satellite SIF has great potential for timely monitoring of heat stress and large-scale assessment of its impact on wheat yield. Since SIF contains information on plant physiological, biochemical and metabolic properties, it is considered a suitable proxy for

vegetation photosynthesis. Previous studies have found that SIF can accurately estimate GPP in different ecosystems and that SIF can also capture changes in GPP due to drought in a timely manner.

In the context of climate change, drought is occurring more and more frequently, and in order to reduce the impact of drought on agriculture, real-time drought monitoring is essential. Currently, drought indices, meteorological parameters and vegetation indices (VIs) are commonly used for drought monitoring [30]. However, they are all inadequate in monitoring the physiological changes of vegetation caused by drought. VIs based on surface reflectance, are more sensitive to the greenness of vegetation. However, the vegetation biomass and canopy structure are not the result of instantaneous photosynthesis, but the photosynthetic yield accumulated over time, so the VIs may not respond to drought in a timely manner [31]. On the other hand, drought indices such as standardized precipitation evaportranspiration index (SPEI) and Palmer drought severity index (PDSI) are calculated using surface water balance based on observed precipitation and temperature data [32]. However, drought indices do not accurately reflect the water stress on vegetation, because plant water effectiveness is also influenced by factors, such as groundwater conditions and soil properties [33]. Soil moisture (SM) is a direct indicator of plant water effectiveness. However, it is a difficult task about how to accurately monitor soil moisture. In contrast, SIF is considered as a direct probe of vegetation photosynthesis, which can capture drought-induced physiological changes in vegetation in a timely manner [34]. Thus, SIF is considered to be a good indicator in monitoring drought. Therefore, the aims of this research were: (1) to design and construct an intelligent irrigation control system utilizing the IoT and (2) to evaluate the responses of SIF and NDVI to different drought conditions.

## 2. Materials and Methods

### 2.1. Experimental Design

The work was performed at Fangshan Comprehensive Experimental Station (39°35′N and 115°42.5′E), Beijing, China, in a warm temperate semi-humid climate (Figure 1) [35]. Rainfall is unevenly distributed throughout the year, with less rainfall in spring and winter and more rainfall in summer and autumn, and the average annual precipitation is 602.5 mm. The soil type of the experimental plot is loam, and the properties are detailed in Table 1.

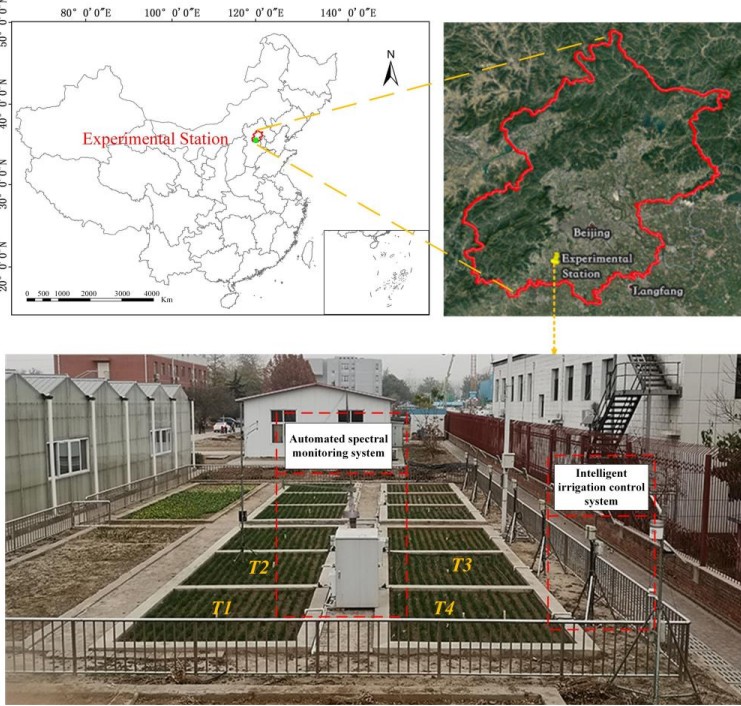

**Figure 1.** The location of experimental station.

**Table 1.** Soil properties in the experimental field.

| Soil Properties | Value |
|---|---|
| soil organic matter | 10.36 g/kg |
| total nitrogen | 0.95 g/kg |
| total phosphorus | 0.28 g/kg |
| clay | 14.5% |
| silt | 40.8% |
| sand | 44.7% |
| field capacity | 25% |
| soil bulk density | 1.39 g/cm$^3$ |

The experiment was carried out in four $3 \times 4$ m plots. Four different water stress gradients were constructed by using the IICS to control the irrigation amount, and three replicates were selected randomly. We controlled the soil moisture of the plots at 60–80% (well-watered, T1), 50–60% (mild water stress, T2), 40–50% (moderate water stress, T3) and 30–40% (severe water stress, T4) of the field water capacity.

### 2.2. Field Management

Winter wheat named Jinnong 7 was planted at the station on 10 October 2018, with a planting density of 52.5 kg/ha$^2$ and rows spaced 20 cm apart. The winter wheat was harvested on 10 June 2019. Chicken manure was fertilized to each experimental plot at 4000 kg/ha$^2$ before the winter wheat planting.

To improve the emergence rate of the winter wheat samples, all of the experimental plots were irrigated with an irrigation amount of 1.05 m$^3$ on 20 November 2018 (41 DAP, days after planting). To ensure the healthy growth of the winter wheat, all of the plots were irrigated at 1.0 m$^3$ on 5 March 2019 (146 DAP).

### 2.3. Description of the Intelligent Irrigation Control System (IICS)

In order to achieve precise irrigation, the IICS based on the IoT was developed (Figure 2). The IICS consists of a soil moisture monitoring system and an automatic irrigation system. It integrates sensors, controllers, actuators and communication equipment. Web-based software can monitor real-time information such as soil moisture, irrigation volume, irrigation times and valve status (open/close).

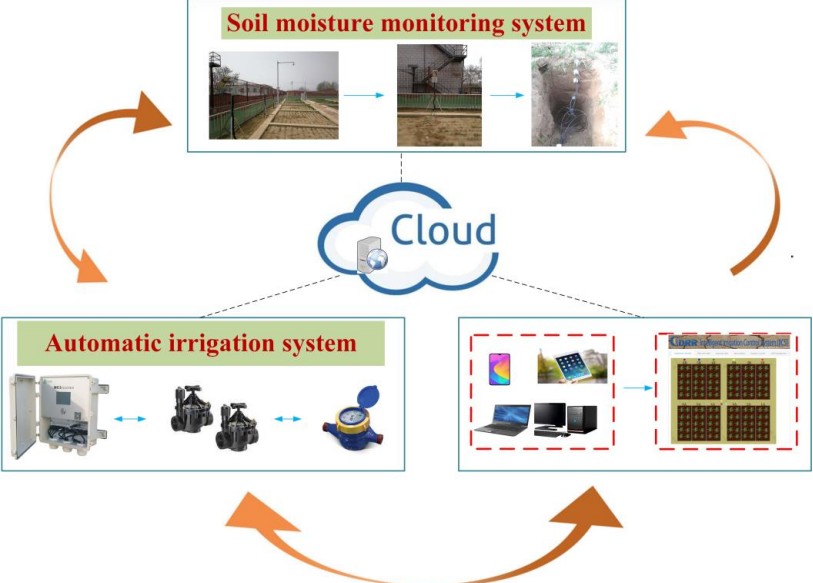

**Figure 2.** Design of intelligent irrigation control system (IICS). The IICS consists of a soil moisture monitoring system and an automatic irrigation system.

### 2.3.1. Soil Moisture Monitoring System

The soil moisture monitoring system consisted of 6 sets of data collectors (MC302 L), 48 Hydra Probe II (Stevens, Portland, Oregon, USA) soil moisture sensors and installation accessories (Figure 3a). The system automatically monitored soil moisture in the 12 plots. Four soil moisture sensors were installed in each plot at depths of 10, 20, 50 and 80 cm below the surface (Figure 3b). Each collector measured data in two control pots through a SDI-12 bus. Data were recorded every 5 min and sent to a cloud server simultaneously. Users can directly log into the website to view, download and analyze the data.

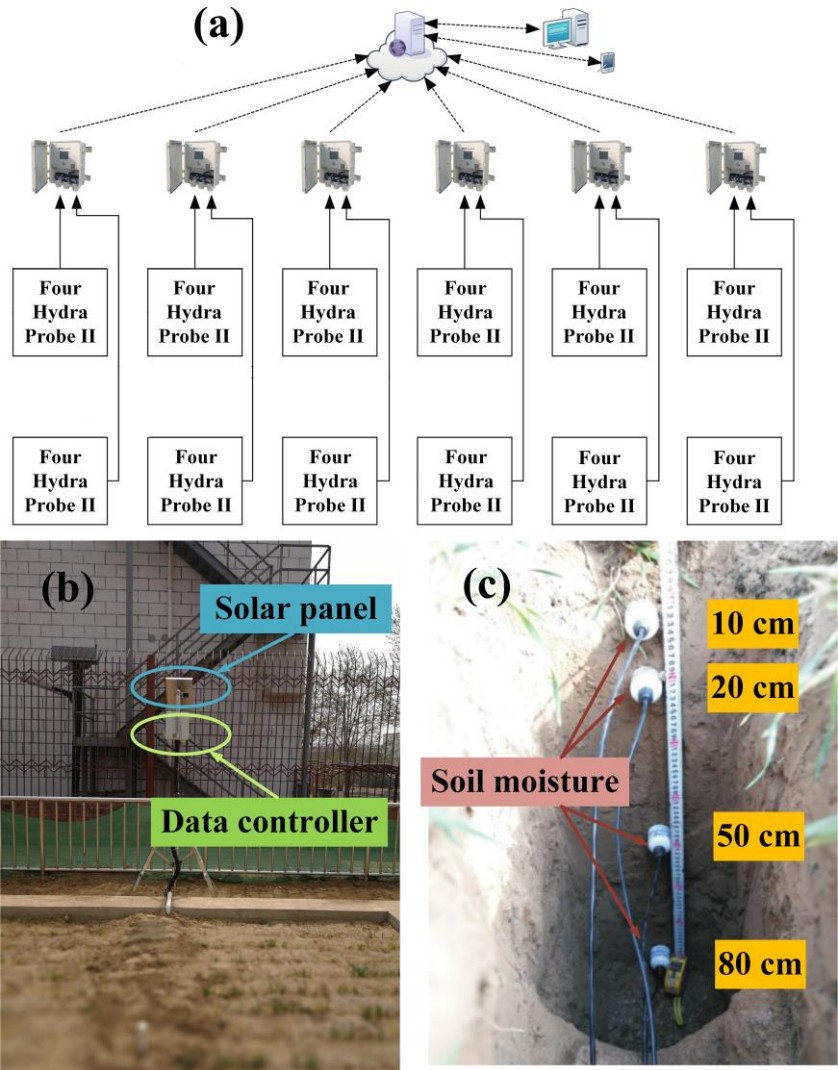

**Figure 3.** (**a**) Soil moisture monitoring system; (**b**,**c**) are the components of the soil moisture monitoring system. It mainly includes solar panel, data collector and soil moisture sensors. The sensors were installed in each plot at depths of 10, 20, 50 and 80 cm below the surface.

Hydra Probe II is an improved version of Hydra Probe soil sensor. The Hydra probe II is calibrated to the soil moisture according to the true dielectric constant, and the calibration is less affected by the soil properties, so the measured data are more accurate. The Hydra Probe II soil sensor supports both SDI-12 and RS-485 communication protocols, providing users with more choices. Different from other soil sensors, the Hydra Probe II soil sensor integrates multiple measurement elements. One measurement can obtain multiple element data such as soil moisture, conductivity, salinity and soil temperature, and it responds quickly, making it easy for users to obtain more information. The sensor can be connected with various types of data collectors and has good compatibility. Hydra Probe II adopts a

compact integrated design and exquisites manufacturing technology to make it suitable for long-term field measurement work without special maintenance.

MC302 L is a low cost, low-power consumption, multifunction data collector with an integrated solar charging controller, polymer lithium battery, GPRS/GPS, true color touch screen and other components. MC302 L can realize measurement and data storage with analogs, switches, frequencies and other interface sensors and is able to send the data remotely through the network.

### 2.3.2. Automatic Irrigation System

The main control element of automatic irrigation system is the programmable logic controller (PLC). PLCs control irrigation according to the set irrigation conditions. The PLC controls the water meter and pulse solenoid valve of each plot. The PLC simultaneously measures the irrigation volume and monitors the state of the solenoid valve (on/off). Automatic irrigation system can monitor data and manage irrigation strategies, agricultural activities, user information, etc. Users can query real-time data, view variable historical data curves, view irrigation records and view irrigation details in the "real-time monitoring" module and add, delete, modify and query agricultural activities in the "farm management" module. Furthermore, users can manage user information in the "system management" module (Figure 4).

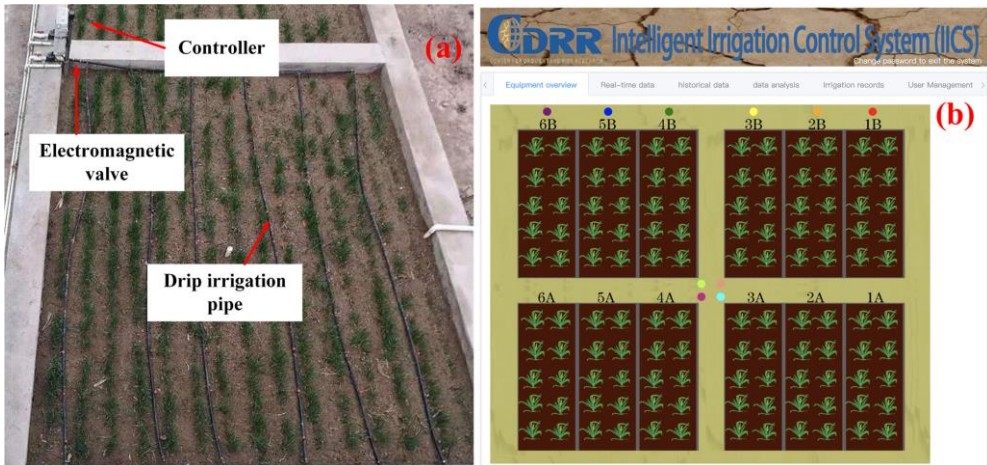

**Figure 4.** Hardware (**a**) and software (**b**) of automatic irrigation system (AIS). Each data collector controls two pots (A and B).

- Real-time monitoring: Data analyses can be performed to view the historical data curve of selected variables within a day, two days interval, a week, a month or a custom time period.
- Irrigation strategy management: Irrigation strategies can be managed, and users can add, delete and modify the starting and ending strategies according to the real conditions. For example, the user can set multiple end conditions, and when any end condition is met, the valve will automatically close.
- View total irrigation records and irrigation details: Users can view the number of irrigation times, irrigation times and irrigation amounts in one day-, two day-, one week- or one-month-long intervals or custom time periods as needed.

### 2.4. Irrigation Scheduling

Measuring the soil moisture is important for optimizing irrigation. The mass balance method was used to calculate the irrigation amounts. The mass balance method, sometimes referred to as scientific irrigation scheduling, is an irrigation schedule determined by calculating how much water is needed based on accurate soil moisture readings and estimates of the

soil properties. Equation (1) can be used to calculate the amount of irrigation needed. The following equation utilizes terms commonly used in soil hydrology studies:

$$Irr = A \times H \times SD \times FC \times \Delta SM, \tag{1}$$

In which Irr ($m^3$) is the irrigation amount, A ($m^2$) is the irrigation area, H (m) is the irrigation depth, SD ($g/cm^3$) is the soil bulk density, FC (%) is the field water capacity and $\Delta SM$ (%) is the difference between the upper and lower limit of the soil moisture threshold.

### 2.5. Measurements of SIF and VIs

The automatic spectral monitoring system was used to collect long-time spectral data in contact (Figure 5). The system consists of QEpro (Ocean Optics, Inc., Largo, FL, USA) spectrometer, electronic switch, optical fiber, cosine corrector and so on. The spectrometer is a QEpro with spectral range of 640~800 nm and spectral resolution of 0.35 nm. One fiber is equipped with a cosine corrector at the end and is used to collect downward solar radiation within a field of view of 180°; the other fiber collects upward radiation reflected from the canopy with a field of view of 25°. These two fibers are fixed to a robotic arm that can be rotated by stepper motors to achieve simultaneous acquisition of spectra between different plots. They are placed in a 23 °C thermostat, which reduces the effect of temperature on the spectrometer. The motor-driven robot arm rotation can continuously collect solar incident spectra and vegetation reflection spectra from four plots.

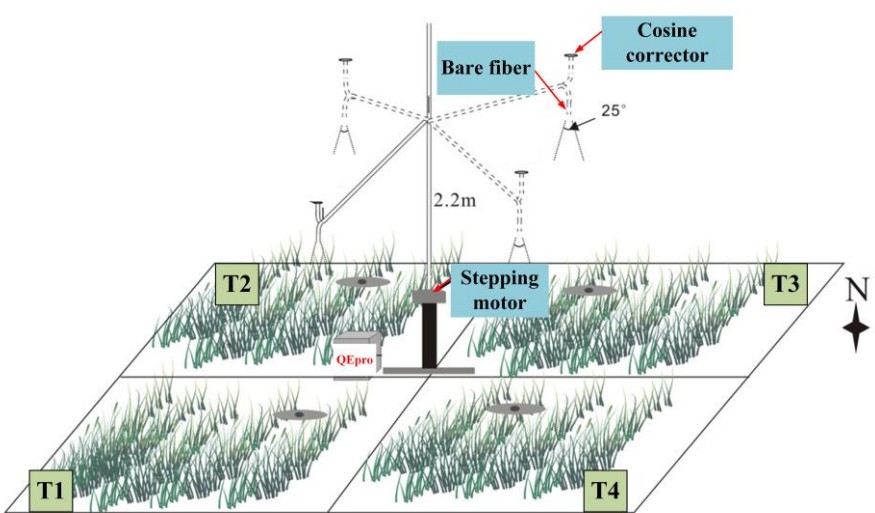

**Figure 5.** The automatic spectral monitoring system. The system consists of QEpro spectrometer, electronic switch, optical fiber, cosine corrector and so on. T1, T2, T3 and T4 represent well-watered, mild water stress, moderate water stress and severe water stress, respectively.

The data processing mainly includes dark current correction, Savitzky–Golay (SG) filter, radiation correction, quality control, extraction of SIF and calculation of VIs. The raw data are dark current corrected to eliminate the effect of dark current on the spectral data. After that, the spectral data are filtered using the SG filter method with a quadratic polynomial of $30 \times 30$ window to eliminate the influence of other noise. The spectral data output from this step are radiation corrected to convert solar radiation and vegetation canopy radiation into irradiance (E, $mW/m^2/nm$) and radiance (L, $mW/m^2/nm/sr$) with energy units, respectively. The SIF is extracted, and the VIs are calculated after quality control.

In this paper, the Spectral fitting method (SFM) is chosen as the extraction algorithm for SIF. The SFM assumes that both fluorescence values and reflectance changes near the Fraunhofer lines can be fitted by mathematical functions. Since SIF has a certain filling effect on the Fraunhofer lines, the SIF at the Fraunhofer lines can be extracted by algorithmic analysis of the measured vegetation canopy upward radiance in the band within the

absorption line and the simulated downward radiance spectrum. The measured canopy upward radiance L(λ) can be expressed as:

$$L(\lambda) = r_{MOD}(\lambda)E(\lambda)/\pi + F_{MOD}(\lambda) + \varepsilon(\lambda) = L_{MOD}(\lambda) + \varepsilon(\lambda), \tag{2}$$

where $r_{MOD}(\lambda)$ is the mathematical function of reflectance used for fitting, $F_{MOD}(\lambda)$ is the mathematical function of fluorescence value used for fitting, $L_{MOD}(\lambda)$ is the radiance simulated by the algorithm and $\varepsilon(\lambda)$ is the difference between the observed and fitted values in each band, representing the error of the model in each band. The parameters of $r_{MOD}(\lambda)$ and $F_{MOD}(\lambda)$ are obtained by solving the system of linear equations using the least squares method to calculate F and r.

The NDVI, NDRE and OSAVI are calculated as follows:

$$NDVI = (R_{750} - R_{685})/(R_{750} + R_{685}), \tag{3}$$

$$NDRE = (R_{790} - R_{720})/(R_{790} + R_{720}), \tag{4}$$

$$OSAVI = 1.16 \times (R_{800} - R_{670})/(R_{800} + R_{670} + 0.16) \tag{5}$$

*2.6. Physiological Measurements*

2.6.1. Relative Water Content (RWC)

The RWC was measured by the weighing method. Six healthy winter wheat samples were selected from each plot randomly; a fully expanded leaf from each winter wheat plant was cut off and quickly weighed to determine the fresh weight of the leaves (G1). Then, the leaves were placed in distilled water and soaked for 6–8 h. The leaves were shaded to avoid photosynthesis. Dry and clean absorbent paper was used to absorb the moisture on the surface of the leaves, and the saturated weight of the leaves was quickly weighed (G2). Then, they were placed in a paper bag, put in an oven after numbering, dried to a constant weight at 85 °C and weighed to measure the dry weight of the leaves (G3). The formula for the RWC is as follows:

$$RWC = (G1 - G3) \times 100/(G2 - G3), \tag{6}$$

2.6.2. Biomass and Yield

Biomass is a characterization of the photosynthetic efficiency of crops and represents the changes in a crop's physiological structure under different water stress treatments. The biomass was mainly collected from the aboveground portion of the winter wheat and measured by the drying method. Ten healthy winter wheat samples were collected from each plot, and dried to constant weight at 80 °C. The weight after processing was the biomass.

After the winter wheat was mature, we randomly selected 1 m$^2$ of winter wheat for harvest in each plot on 10 June. The grain yield was weighed after threshing and drying. The 1000-grain weight was obtained by weighing a thousand grains.

*2.7. Statistical Analysis*

Analysis of variance (ANOVA) was implemented using statistical software SPSS21 (IBM, Armonk, NY, USA). Differences between means were detected using the LSD test with a significance level of $p < 0.05$.

**3. Results**

*3.1. Response of the Automated Irrigation Scheduling*

Different water stress treatments began on 4 April 2019 (176 DAP). According to the subsequent irrigation strategies, T1 was irrigated automatically on 176 DAP, 187 DAP, 192 DAP, 203 DAP, 207 DAP, 209 DAP, 212 DAP and 215 DAP, with a total irrigation amount of 1.8 m$^3$, while T4 was irrigated only once on 11 May, with an irrigation amount of only 0.2 m$^3$. T2 and T3 were irrigated four times and three times, respectively, with irrigation

amounts of 1.1 m³ and 0.7 m³, respectively. Compared with T1, the irrigation amounts of T2, T3 and T4 decreased by 18.18%, 28.57% and 41.56%, respectively. The irrigation times and irrigation amounts of experimental plots under different water stresses are shown in Table 2.

**Table 2.** Irrigation times and irrigation amounts of experimental plots under different water stresses.

| Plot | No. of Irrigations | Irrigation Amount (m³) |
| --- | --- | --- |
| T1 | 9 | 3.85 |
| T2 | 6 | 3.15 |
| T3 | 4 | 2.75 |
| T4 | 3 | 2.25 |

Since the roots of winter wheat were densely distributed at 0–50 cm, the soil moisture in this study was represented by the average soil moisture at depths of 10, 20 and 50 cm. Before the water stress treatments, the soil moisture values of each plot were similar. When the irrigation strategies of each plot were establishing using an IICS based on the IoT, automatic irrigation was implemented, and the soil moisture of each plot showed varied trends. On 176 DAP, as the soil moisture of plots T1 and T2 reached the lower threshold of irrigation conditions, the IICS irrigated the two plots and their soil moisture increased. The soil moisture of plots T3 and T4 plots had not yet reached the threshold for irrigation conditions, so their soil moisture continued to decline, and irrigation was not initialized until the threshold of irrigation conditions was reached. We can know that the soil moisture of T1 fluctuated more frequently (Figure 6). This was because the threshold of the soil moisture for irrigation conditions must be set high to achieve a drought-free state, more irrigation events are triggered. However, T4 was set to receive the severe water stress treatment, and the threshold of the soil moisture for irrigation conditions was set very low; therefore, only one irrigation was carried out on 213 DAP. In the other cases, the fluctuations in the soil moisture values were caused by precipitation.

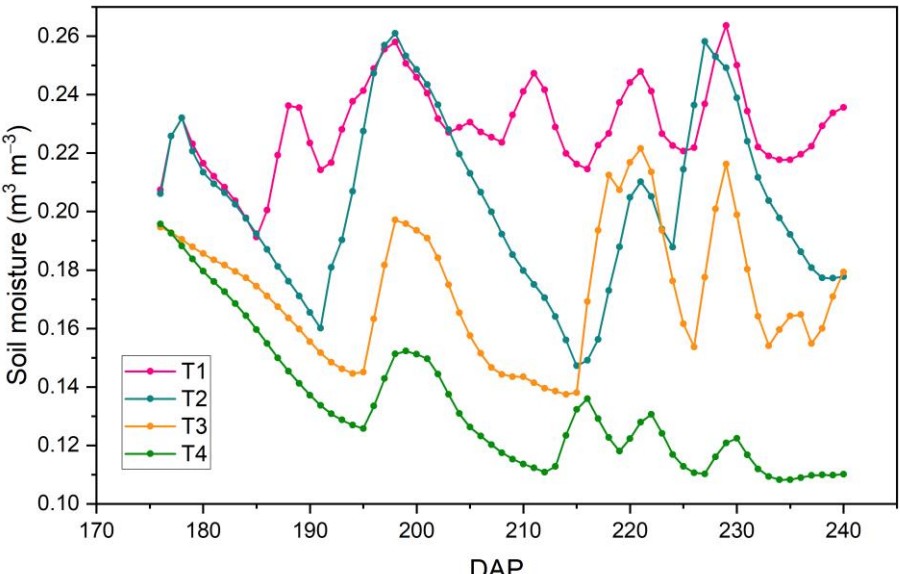

**Figure 6.** Seasonal variation of soil moisture under different water stresses. The soil moisture was represented by the average soil moisture at depths of 10, 20 and 50 cm. DAP means days after planting.

After receiving the water stress treatment, the soil moisture of each plot was maintained under user-defined conditions, expressed as T1 > T2 > T3 > T4, which showed that the IICS was able to realize automatic irrigation and control the soil moisture at an ideal level (Figure 7).

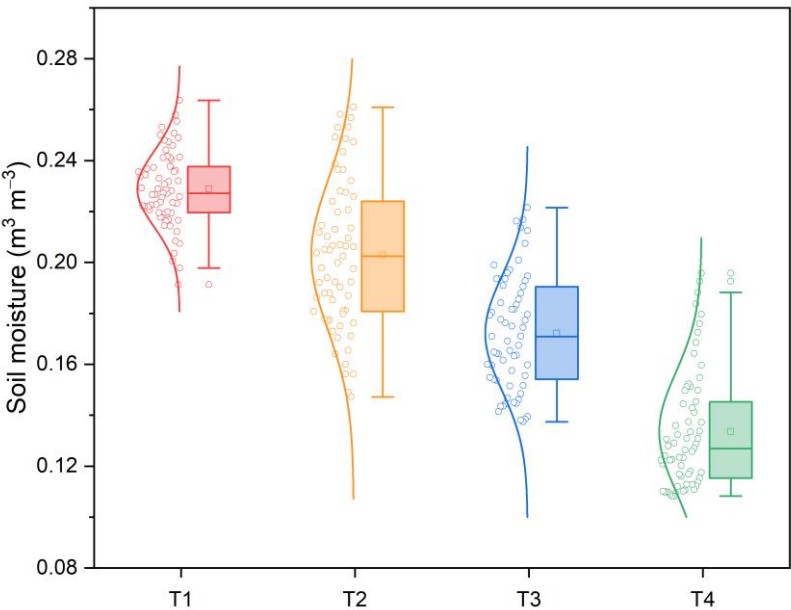

**Figure 7.** Comparison of the average soil moisture under different water stresses. The soil moisture was represented by the average soil moisture at depths of 10, 20 and 50 cm. The average was calculated using the values of soil moisture from 176 DAP to 240 DAP. DAP means days after planting.

In this study, the T1 plot was selected to explore the variation characteristics of soil moisture in different layers. The trends of soil moisture values at 10 cm and 20 cm were similar, and their fluctuations were relatively severe (Figure 8). The soil moisture changes observed at 50 cm and 80 cm were relatively gentle. This was because the soil sensors at 10 cm and 20 cm were close to the ground surface and were greatly affected by irrigation. The soil moisture at 20 cm was the highest value observed, because it was greatly affected by irrigation and was less affected by evapotranspiration. The soil moisture at 50 cm was the lowest and was even lower than that at 80 cm. This was because, although the soil moisture values at 50 cm and 80 cm were not greatly affected by irrigation, the soil moisture at 80 cm was less affected by evapotranspiration.

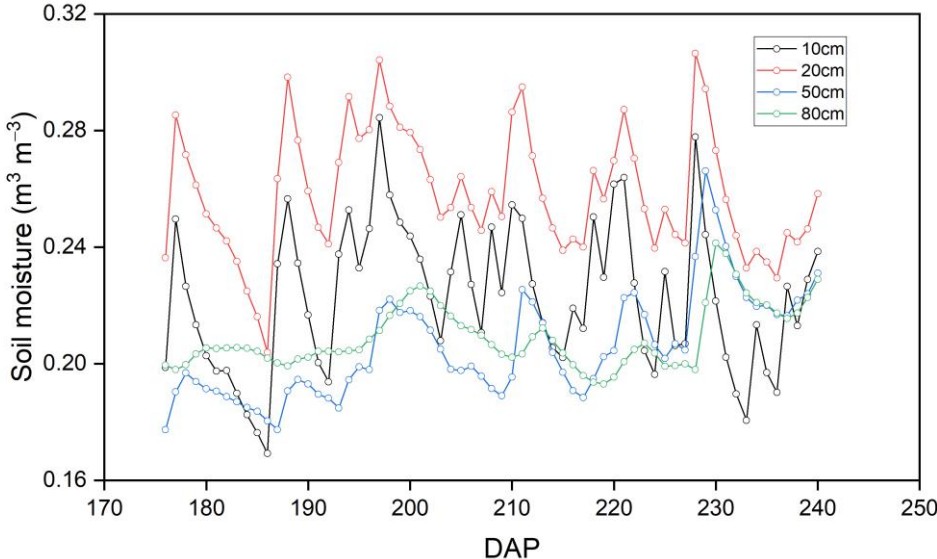

**Figure 8.** Seasonal variation of soil moisture at different depths in T1. DAP means days after planting.

### 3.2. Physiological Results

3.2.1. Relative Water Content

The RWC is an indicator of water stress. The larger the value is, the higher the water content is. From Figure 9, it can be seen that the RWC of each plot showed a slowly decreasing trend. This was due to the gradual senescence of the leaves as the winter wheat grew; the water-holding capacity of the leaves therefore decreased. During the growing season, the RWC showed an overall trend of T1 > T2 > T3 > T4, which resulted from different soil moisture. This also suggested that the construction of the water stress gradient was successful.

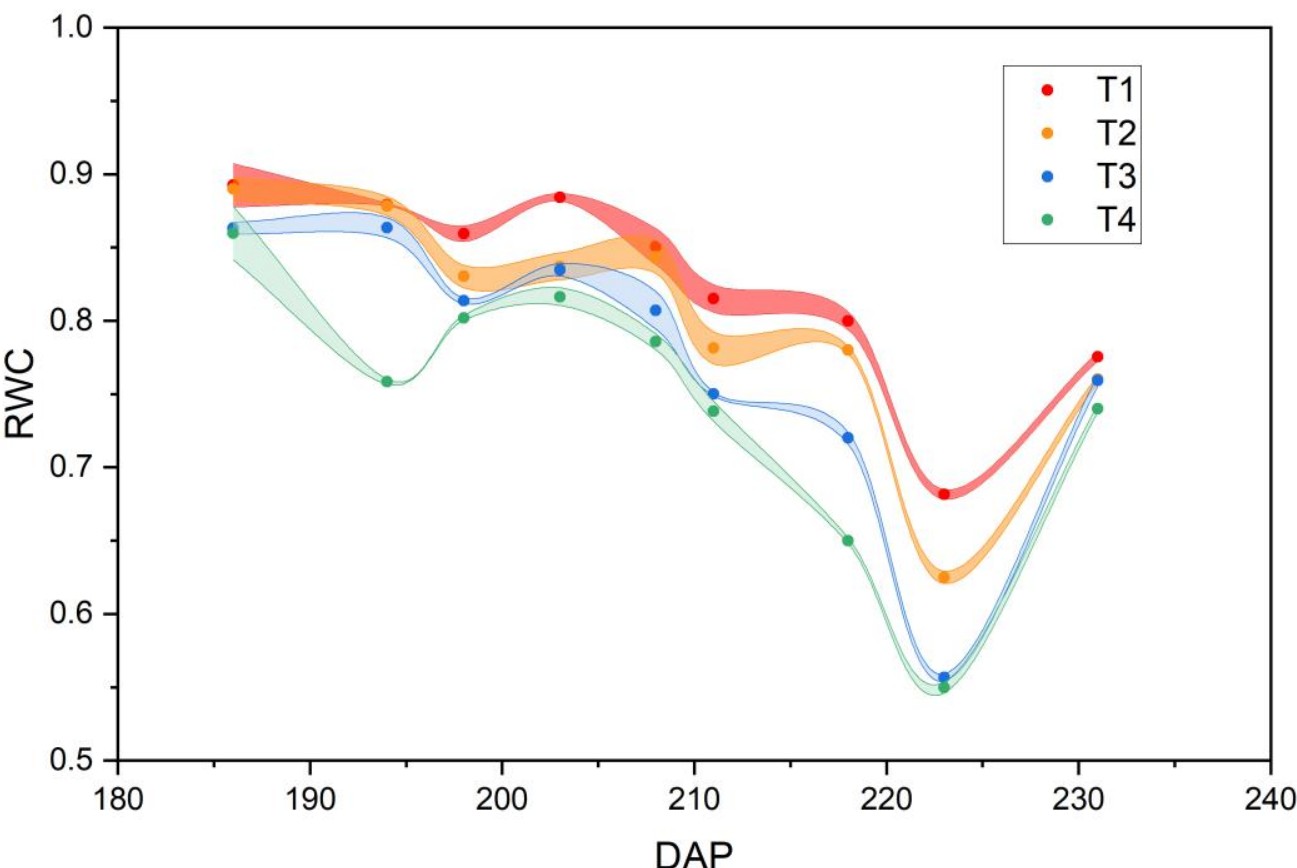

**Figure 9.** Seasonal variations of the relative water content (RWC) under different water stresses. Each point and bar indicate the mean value ± standard deviation (SD). DAP means days after planting.

3.2.2. Biomass and Yield

This study showed that the aboveground biomass of winter wheat varied with different water stress levels (Figure 10). From jointing to maturity, the biomass produced by plot T1 was significantly higher than that of plots T3 and T4.

In addition, the 1000-grain weights of winter wheat were significantly reduced in T3 and T4, while there was no significant change in T2 (Table 3). Relative to T1, the 1000-grain weights of T3 and T4 decreased 23.22% and 14.28%, respectively. The grain weights of the winter wheat samples were affected by water stress. The grain weight of T2 only decreased by 7.27%, while the grain weights of T2 and T3 decreased by 32.73% and 43.64%, respectively.

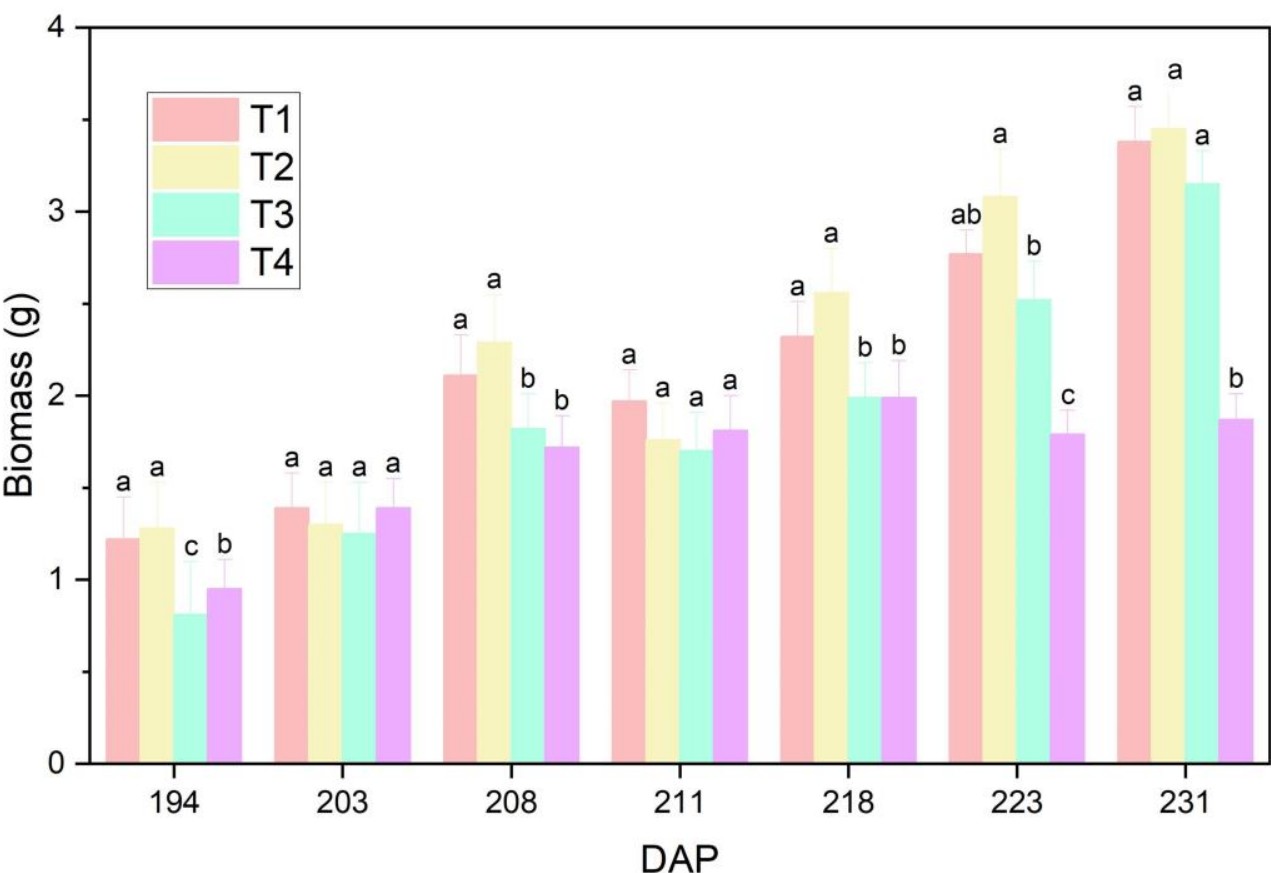

**Figure 10.** Seasonal variations of biomass under different water stresses. Each point and bar indicate the mean value ± standard deviation (SD). DAP means days after planting. Values with different letters indicate significant differences at *p* < 0.05.

**Table 3.** Irrigation times and irrigation amounts of experimental plots under different water stresses. Each value indicates the mean ± standard deviation (SD). In a column, values with different letters indicate significant differences at *p* < 0.05.

| Plot | 1000-Grain Weight (g) | Grain Weight (kg/m$^2$) |
|:---:|:---:|:---:|
| T1 | 41.27 ± 1.71 a | 0.55 ± 0.02 a |
| T2 | 42.08 ± 2.46 a | 0.51 ± 0.04 a |
| T3 | 35.38 ± 1.64 b | 0.37 ± 0.02 b |
| T4 | 31.69 ± 2.42 b | 0.31 ± 0.03 c |

*3.3. Responses of SIF and VIsto Water Stress*

To explore the response of SIF, NDVI, NDRE and OSAVI to different water stresses, we calculated the mean values of SIF, NDVI, NDRE, OSAVI and SM from 5 April to 21 May. As shown in the Figure 11, different water stresses caused different effects on SIF and VIs, while the responses of the NDRE, OSAVI and NDVI to water stress were similar. By one-way ANOVA, SIF decreased significantly under mild drought, while the NDVI, NDRE and OSAVI did not change significantly under mild drought and decreased significantly only under moderate and severe drought. This indicated that SIF is more sensitive to drought.

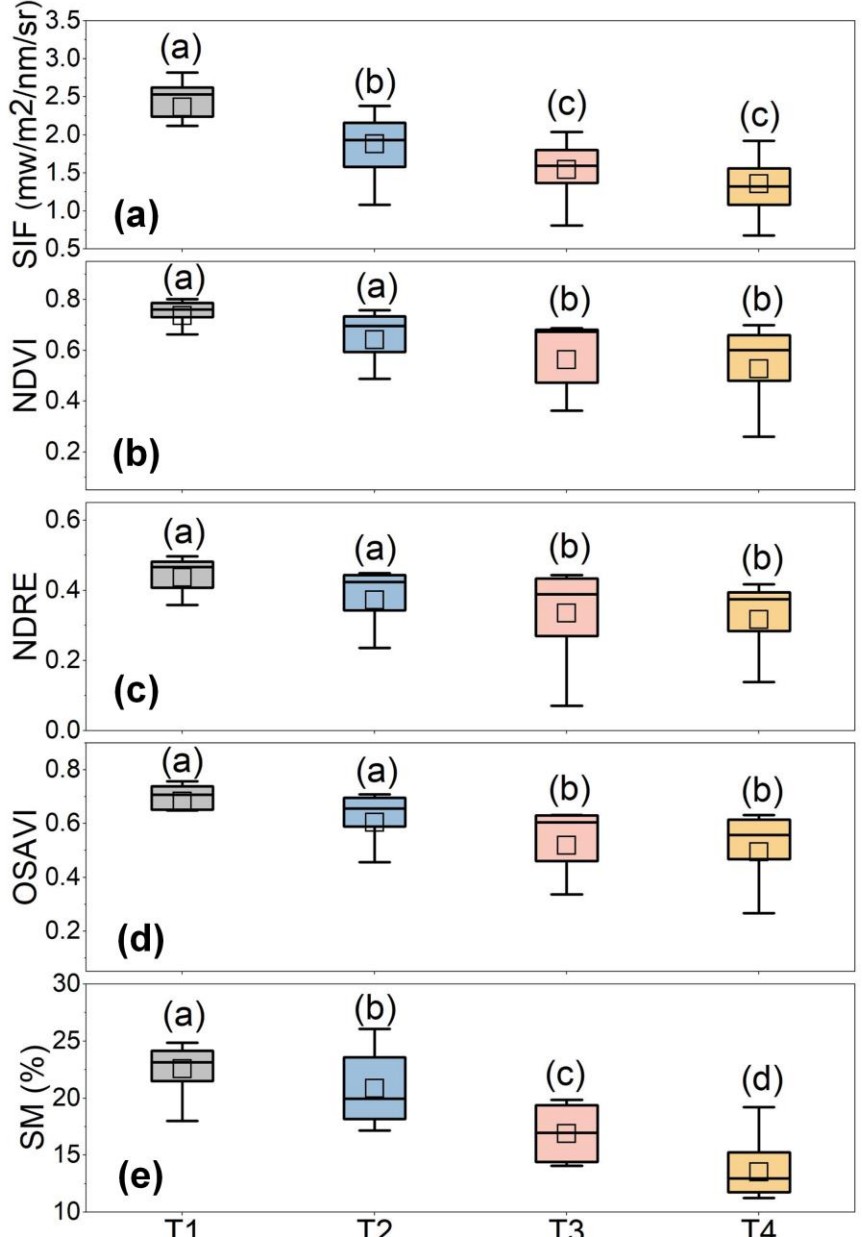

**Figure 11.** The responds of (**a**) SIF, (**b**) the NDVI, (**c**) NDRE, (**d**) OSAVI and (**e**) SM under different water stresses. Values with different letters indicate significant differences at $p < 0.05$. The hollow blocks in the figure represent the average values, which were calculated using the data collected from 177 DAP to 223 DAP.

To investigate the response of SIF and VIs to irrigation and precipitation in depth, we also analyzed the variation characteristics of SIF and VIs during the growing season. The NDVI, NDRE and OSAVI were relatively stable during the growing season, showing a trend of increasing and then decreasing, while SIF showed fluctuating changes, mainly due to the influence of irrigation or precipitation (Figure 12). The SIF showed a decreasing trend on 186 DAP and a sudden increase on 191 DAP, followed by a decreasing trend until the next irrigation, while the SIF showed similar changes on 199 DAP and 203 DAP. The NDVI, NDRE and OSAVI did not show frequent fluctuations, and only plot T1 showed fluctuations on 191 DAP. The above results indicate that SIF is sensitive to soil moisture and can respond to changes in the soil moisture after irrigation.

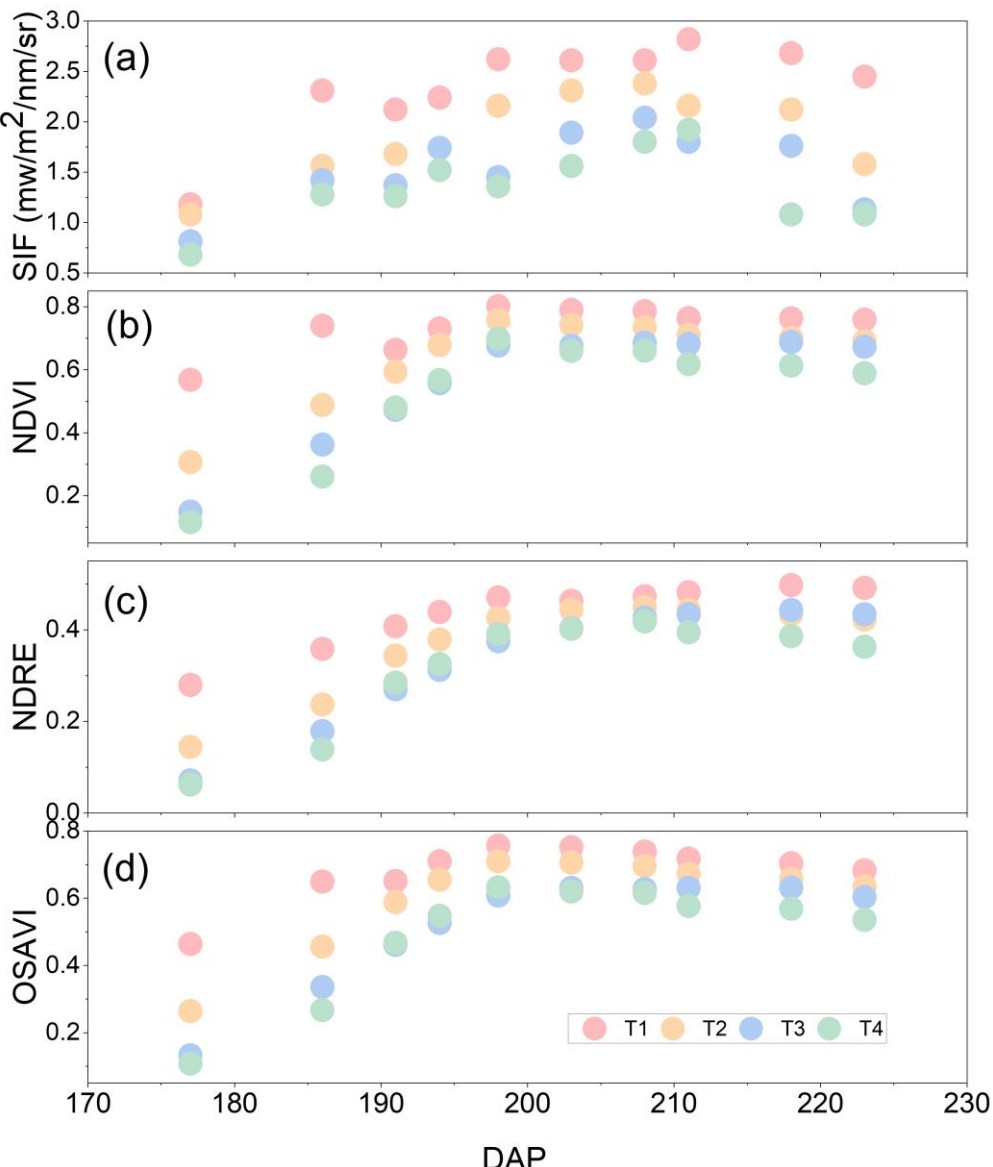

**Figure 12.** The seasonal changes of (**a**) SIF, (**b**) NDVI, (**c**) NDRE and (**d**) OSAVI under different water stresses. The red, yellow, blue and green solid circles represent well-watered, mild stress, moderate stress and severe stress, respectively. All values are averaged from 9:00 to 16:00. DAP means days after planting.

### 3.4. Cost Analysis

Table 4 shows the list and description of projects used to design and implement irrigation control and automatic spectral acquisition equipment. The cost analysis is expressed in U.S. dollars (USD). Due to the fluctuation of the exchange rate in the global market, these costs will also change accordingly. The costs shown here do not include transportation costs and labor costs. It can be seen from Table 4 that the cost of the IICS is $35,111.8 USD, and the cost of the automatic spectral monitoring system is $19,429 USD, totaling $54,540.8 USD. We think this set of equipment is expensive and not suitable for farmers to use directly. The most expensive of this set of equipment is the soil moisture sensors and spectrometer, with the prices of $33,472.8 USD and $17,433.75 USD, respectively. As we are conducting basic scientific research, considering the accuracy requirements, we use the instruments with the highest accuracy produced in the United States, which directly increases the cost of investment. Nowadays, some domestic equipment can meet

the requirements of agricultural production. Most importantly, domestic equipment has greatly reduced the cost and made it affordable for farmers.

**Table 4.** Cost analysis of the Intelligent Irrigation Control System and Automatic Spectral Monitoring System.

| NO. | Category | Item Description | Unit Quantity | Unit Price ($) | Amount ($) |
|---|---|---|---|---|---|
| 1. | | Data collectors (MC302 L) | 6 | 20.25 | 121.50 |
| 2. | | Soil moisture sensors (Hydra Probe II) | 48 | 697.35 | 33,472.80 |
| 3. | | Mounting brackets | 6 | 25.86 | 155.16 |
| 4. | | Solar panels | 6 | 10.63 | 63.78 |
| 5. | Intelligent irrigation control system | Electrolytic capacitor | 6 | 5.89 | 35.34 |
| 6. | | Resistors | 6 | 8.62 | 51.72 |
| 7. | | 12-V Relay Module External Trigger Delay Adjustable | 6 | 7.59 | 45.54 |
| 8. | | Module Light Emitting Diode | 4 | 3.46 | 13.84 |
| 9. | | Digital Temperature, Humidity Sensor Module | 4 | 7.84 | 31.36 |
| 10. | | Water Meter | 12 | 4.28 | 51.36 |
| 11. | | Air temperature and humidity sensor | 4 | 267.35 | 1069.40 |
| | Subtotal 1 | | 108 | | 35,111.80 |
| 12. | | QEpro spectrometer | 1 | 17,433.75 | 17,433.75 |
| 13. | | electronic switch | 1 | 285.71 | 285.71 |
| 14. | Automatic spectral monitoring system | optical fiber | 2 | 714.29 | 1428.58 |
| 15. | | cosine corrector | 1 | 166.67 | 166.67 |
| 16. | | Robotic arm | 1 | 71.43 | 71.43 |
| 17. | | Stepping motor | 1 | 42.86 | 42.86 |
| | Subtotal 2 | | 7 | | 19,429.00 |
| | Total | | 115 | | 54,540.80 |

## 4. Discussion

### 4.1. Mechanisms of the Drought on SIF and Physiological Parameters

From jointing to maturity, the biomass produced by plot T1 was significantly higher than that of plots T3 and T4 (Figure 10). These findings were consistent with previous studies, which reported reductions in biomass under water stress. Under moderate and severe stresses, the aboveground biomass decreased significantly, which was attributed to the decreased photosynthetic activities caused by water deficits. As a result, the growth of leaves was affected and could not be fully extended. The intercepted photosynthetic active radiation (PAR) of winter wheat was reduced, leading to a decrease in the plant height and substance assimilation accumulation, resulting in a decrease in the biomass. However, when compared with T1, the aboveground biomass of T2 was not significantly decreased. This could be because photosynthetic activities were not largely affected under mild stresses. The 1000-grain weights of T3 and T4 decreased due to the decreased grain filling time, which resulted in a shorter duration and a lower dry matter accumulation. Thus, the seeds in T3 and T4 developed less, which led to a lower 1000-grain weight. It was interesting that the 1000-grain weight of T2 increased by 1.95% compared to T1, which could be attributed to the fact that the photosynthesis of the winter wheat plants in T2 was not weakened [36]. The grain weight of T2 only decreased by 7.27%, while the grain weights of T2 and T3 decreased by 32.73% and 43.64%, respectively (Table 3). This was similar to the results of Ref. [37], who noted that crop yields decrease under water stress conditions. This study suggested that water stress conditions may reduce the yield by decreasing the 1000-grain weight. As we all know, moderate and severe water stress conditions significantly shortened the duration of carbon assimilation, reduced the photosynthesis rate and affected the transformation of photosynthetic products to the yield, which resulted in a reduction in the yield of winter wheat in plots T3 and T4. Winter wheat maintained a higher photosynthetic rate in T2, which was conducive to full grain filling. Therefore, the yield of T2 was not reduced significantly, which mirrored previous research results [38].

When drought occurs, the decrease in SIF is mainly due to changes in chlorophyll fluorescence quantum yield and absorbed photosynthetically active radiation (APAR) [39]. Drought causes stomatal closure of vegetation, which reduces $CO_2$ uptake and leads to a

decrease in vegetation photosynthetic rate. At the same time, drought decreases the leaf area index (LAI) of vegetation, which causes a corresponding decrease in the fraction of photosynthetically active radiation (FPAR) and ultimately leads to a decrease in vegetation photosynthesis and chlorophyll fluorescence excitation energy [40]. The APAR absorbed by vegetation is mainly used for photosynthesis, heat dissipation and emitted fluorescence. There is a competitive relationship between them, but in fact, the relationship between their three is more complex and not unique. For example, the relationships between fluorescence and photochemical reactions were negatively and positively correlated under low and high light stresses, respectively [41]. Although drought reduces the proportion of APAR used for photosynthesis, the fluorescence quantum yield changes are more complex due to the complex heat dissipation mechanism of vegetation [42]. Therefore, the effect of drought on SIF needs further in-depth study. However, it is clear that, when drought occurs, the photosynthetic rate of vegetation decreases significantly (Table 3). SIF has also been shown to have the ability to monitor the decline of photosynthesis in plants.

### 4.2. Potential Application of SIF in Drought Monitoring

In this study, SIF and VIs of winter wheat were monitored over a long period of time during the growing season to investigate the differences in SIF and VI responses to different water stresses. This study will provide valuable support for the application of SIF in drought monitoring.

Through ANOVA, we found that SIF was sensitive enough to light drought, while NDVI, NDRE and OSAVI was significantly reduced only in moderate and severe drought (Figure 11). This is consistent with the findings of a previous study [27]. They compared the effects of drought on SIF and the Enhanced Vegetation Index (EVI) of Amazonian forests, and they noted that SIF in Central Amazonia significantly decreased during the dry season, while the EVI showed relatively little change. A shortcoming of traditional Vis is that they reflect changes in the chlorophyll content rather than directly monitoring the photosynthesis of vegetation. In contrast, SIF is considered to be a direct probe of photosynthesis and therefore more suitable for drought monitoring.

Some studies have pointed out that, when the LAI exceeds 5, the NDVI will be saturated, and it will not be able to capture changes in vegetation [43]. Therefore, the NDVI is not effective for drought monitoring during periods of dense vegetation growth. The NDRE can be used to monitor the growth of crops at the mature stage, while the OSAVI is more effective in identifying the chlorophyll content of plants at the early growth stage. However, our research found that the NDRE and OSAVI also could not monitor agricultural drought in time. Since SIF and vegetation photosynthesis are closely related, SIF still has the ability to accurately monitor vegetation changes even when the LAI is high and vegetation canopy cover is high. Therefore, compared with the NDVI, NDRE and OSAVI, SIF is more suitable for drought monitoring in areas with high vegetation cover.

By analyzing the seasonal variation patterns of SIF, the NDVI, NDRE and OSAVI, it can be found that SIF responds to drought more quickly, while the NDVI, NDRE and OSAVI only respond to drought at a longer time scale (Figure 12). This indicates that SIF can monitor the occurrence of drought quickly, while the NDVI, NDRE and OSAVI are suitable for monitoring drought over a long period of time. This is consistent with the findings of Liu et al. [28]. Liu et al. [28] showed that the correlation between F760/PAR and the SM was better for shorter time lags but not significant at longer time lags compared to the NDVI. The late response of the NDVI to changes in the SM may be due to the influence of changes in the chlorophyll content. Furthermore, background signals such as soil color and shading also have significant effects on the NDVI [44]. In addition, the vegetation type also affects the response of SIF, the NDVI, NDRE and OSAVI to drought. Unlike vegetation types such as crops and grasslands, SIF, the NDVI, NDRE and OSAVI of mixed forests did not show significant decreases under drought conditions [45]. In summary, the NDVI, NDRE and OSAVI cannot track the occurrence of early drought in a timely manner, while

SIF can capture the characteristics of drought-induced changes in vegetation physiology in real time and is a valuable indicator for agricultural management and disaster monitoring.

### 4.3. Limitations and Further Directions

In this study, the applicability of SIF obtained from field experiments in drought monitoring was explored, and good results were obtained. However, to upscale SIF for satellite remote sensing, a lot of disturbance factors need to be considered, such as the soil background. With the development of sensor technology and the continuous improvement of fluorescence extraction algorithm, several new instruments on geosynchronous satellites, including OCO-3 and Sentinel-4, have been launched or are planned to be launched in the next few years [46]. At that time, SIF products with higher spatial resolution and data quality will become a reality. This will greatly promote the research of drought monitoring, vegetation productivity and terrestrial carbon cycle.

At present, the resolution of satellite SIF products used is relatively coarse. Researchers have used downscaling methods to obtain continuous high spatial and temporal resolution SIF data [42]. In addition, the deep learning method has been widely used [47]. In the future, the deep learning method will be used to downscale the SIF data and to establish a drought index incorporating SIF to improve the accuracy of drought monitoring.

SIF is the most direct signal used to detect photosynthesis in a terrestrial vegetation ecosystem, which can accurately capture the response of the vegetation physiological state to environmental changes. The coarse spatial and temporal resolution of SIF limits its ability to detect photosynthetic activity in terrestrial vegetation. The relationship between SIF and photosynthesis differs under the influence of environmental changes, e.g., when water deficit conditions occur. The nonlinear relationship between SIF and photochemical reactions exists not only at the leaf scale but also at the canopy scale. The complex relationship between SIF and photosynthesis needs further investigation.

### 5. Conclusions

An IoT-based IICS was designed and built in this study. This system can monitor the soil moisture levels of different profile layers in real time and perform calculations and analyses to realize automatic irrigation. The IICS can be controlled remotely and can involve other professionals to help users make better decisions. The system also enables various controlled irrigation treatments. In this study, we successfully constructed different regulated deficit irrigations through the IICS utilizing the IoT. The soil moisture of each plot was accurately controlled within the set range. The influential characteristics of SIF, the NDVI, NDRE and OSAVI of winter wheat on different regulated deficit irrigations were analyzed. The effects of different drought intensities on SIF, the NDVI, NDRE and OSAVI were different. Compared to well-watered, mild, moderate and severe drought, all had significant effects on SIF. In contrast, the NDVI, NDRE and OSAVI did not change significantly under mild drought and decreased significantly only under moderate and severe droughts. Moreover, the response of SIF was more drastic than the NDVI, NDRE and OSAVI when irrigation was performed. The results indicate that SIF is more sensitive to drought than the NDVI, NDRE and OSAVI and more suitable for drought monitoring.

**Author Contributions:** Conceptualization, W.Z.; methodology, W.Z.; software, W.Z.; validation, W.Z. and Q.S.; formal analysis, W.Z.; investigation, W.Z., J.Y., X.H. and Q.S.; resources, J.W.; data curation, W.Z.; writing—original draft preparation, W.Z.; writing—review and editing, W.Z.; visualization, W.Z.; supervision, J.W.; project administration, J.W.; and funding acquisition, J.W. All authors have read and agreed to the published version of the manuscript.

**Funding:** This research was funded by the National Natural Science Foundation of China, grant number 42071401.

**Data Availability Statement:** The data presented in this study are available on request from the corresponding author.

**Acknowledgments:** The authors gratefully acknowledge Mei Liu for providing valuable suggestions on the modification of the pictures.

**Conflicts of Interest:** The authors declare no conflict of interest.

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
