# Peer review of "Exploring the Ability of Solar-Induced Chlorophyll Fluorescence for Drought Monitoring Based on an Intelligent Irrigation Control System"

_remotesensing, doi:10.3390/rs14236157_

Round 1
Reviewer 1 Report
This is a well-organized study, with meaningful results and conclusions, which evaluate the utility of solar-induced chlorophyll fluorescence for drought monitoring of winter wheat. In particular, this study shows us advanced intelligent irrigation control system and automatic spectral monitoring system, which can support the authors to do a lot of valuable research. This study is innovative but there are still some issues should be addressed before being accepted for publication.
1. In the abstract, the existing problems in the current research need to be added. You can use 1-2 sentences to explain the shortcomings and the significance of the study.
2. Line 25, “regulated deficit irrigation (RDI)”. Abbreviations should be defined in parentheses the first time they appear in the abstract if the term appears more than once. RDI appears only once in the abstract, so there is no need for abbreviation.
3. This study focuses on exploring the ability of SIF for drought monitoring. So, you should add the research progress of SIF in the Introduction. For example, the application of SIF in stress (drought, heat wave) monitoring, estimation of GPP, etc.
4. Line 130-140, the irrigation times and irrigation amounts of experimental plots under different water stresses were shown. These were the results obtained from the intelligent irrigation control system, and I think this part is more appropriately placed in Section 3.1.
5. Line 152 and 216. When we describe a device, we should use the following format. Manufacture, city, state acronym, country.
6. In section 2.5, the automatic spectral monitoring system is advanced and valuable, and I propose to describe this system in detail. For example, the components of the system, the band range and resolution of QEpro, etc.
7. In section 2.5, the processing of spectral data should be supplemented, which can be a reference for other researchers.
8. Line 219, “Spectral fitting method (SFM) was used to extract SIF.” There are many algorithms to extract SIF. I recommend to introduce the extraction algorithm SFM for SIF in detail.
9. In the captions of Figures 6, 11 and 12, they should be stated which data were used to calculate the results. Such as, all measurements in the figure were collected from ** to **. SIF was calculated from the measurements from ** to **.
10. SIF and photosynthesis of vegetation are closely related, which is an advantage of SIF to monitor drought. In lines 367 and 375, you mention that drought can reduce photosynthesis of vegetation. I suggest adding a graph here to show the effect of drought on the photosynthetic rate of vegetation, which would provide strong support for this study.
11. This study used canopy SIF observed by QEpro above the ground. To upscale SIF to satellite remote sensing, a lot of disturbance factors need to be considered, such as the soil background, the illumination geometry, etc. Please include a broader discussion and applicability.
12. The limitations of this study and what could be done in future studies should also be described in Discussion. For example, SIF was obtained using an automated spectral monitoring system. SIF is closely related to photosynthesis of vegetation, and I guess it should be related to the phenology of vegetation as well. And the irrigation strategy is related to the phenology. I suggest that the use of SIF for precise irrigation management is a hot spot in the future.
Reviewer 2 Report
Congratulations to the authors of the manuscript. The idea behind the drought monitoring system is fantastic. However, I have two main concerns regarding the manuscript and the analysis conducted. First, how much would cost to implement such a system in a commercial field? The authors discussed the advantages and disadvantages of conventional methods for irrigation management, and at the same time stated throughout the paper that the SIF is a better option for drought monitoring and irrigation management. Thus, I wonder if there is a possibility of adding a simple cost analysis only to describe how feasible would be for a farmer to adopt such technology. Second, I would like to see other VIs than the NDVI in this study. The authors clearly stated in lines 395 – 405 that the NDVI was not a good option to discriminate the water stress levels. The NDVI is known to present saturation issues at certain crop stages. Based on that, I believe that this study could benefit from adding other VIs (ex: NDRE, MCARI1, OSAVI, etc). This would make the results more robust and show that the SIF is a better option than any VI.
